# Impact of Macronutrient Fertility on Mineral Uptake and Growth of *Lactuca sativa* 'Salanova Green' in a Hydroponic System

Patrick Veazie [1,*], Piyush Pandey [2], Sierra Young [2,†], M. Seth Ballance [1], Kristin Hicks [3] and Brian Whipker [1,*]

1   Department of Horticultural Sciences, North Carolina State University, Raleigh, NC 27695, USA
2   Department of Biological and Agricultural Engineering, North Carolina State University, Raleigh, NC 27695, USA
3   North Carolina Department of Agriculture and Consumer Services, Raleigh, NC 27699, USA
*   Correspondence: phveazie@ncsu.edu (P.V.); bwhipker@ncsu.edu (B.W.)
†   Current address: Department of Civil and Environmental Engineering, Utah State University, Logan, UT 84322, USA.

**Abstract:** *Lactuca sativa* (commonly referred to as lettuce) is one of the most popular grown hydroponic crops. While other fertilizer rate work has been conducted on lettuce, the impact of each element has not been evaluated independently or by determining adequate foliar tissue concentrations when all nutrients are plant-available. This study explores the impact that macronutrients have on the growth and yield of lettuce at different stages of the production cycle. Additionally, this study explores the adequate nutrient rates by regressing nutrient curves to find the concentration of each element that corresponds to optimal growth. Plants were grown under varying macronutrient concentrations (0, 8, 16, 32, 64, and 100%) utilizing the concentrations of a modified Hoagland's solution based on 150 mg·L$^{-1}$ N. Lettuce plants were grown in a silica sand culture and received a nutrient solution in which a single element was altered. Visual symptomology was documented, and leaf tissue mineral nutrient concentrations and biomass were measured at Weeks 3, 6, and 8 after transplant. Optimal elemental leaf tissue concentration and biomass varied by macronutrient rates and weeks of growth. Nitrogen rate produced a linear increase in total plant dry weight, but foliar N followed a quadratic plateau pattern. Other elements, such as phosphorus, potassium, and magnesium, produced distinct total plant dry weight plateaus despite increasing fertility concentrations. These results demonstrate that fertility recommendation can be lowered for nutrients where higher rates do not result in higher plant biomass or foliar nutrient concentrations.

**Keywords:** lettuce; nutrient rates; fertility; nitrogen; phosphorus; potassium; calcium; magnesium; sulfur

## 1. Introduction

Lettuce, *Lactuca sativa*, is a cool-season vegetable that is produced in the field during the fall and spring of the southern United States when temperatures range between 7 and 24 °C [1]. However, the increasing consumer demand for high quality and off-season availability has increased greenhouse production [2]. Lettuce is the most commonly grown plant in aquaponic and hydroponic systems [3]. Lettuce has four common morphological forms: loose-leaf, butterhead, crisphead, and romaine [4]. However, butterhead and loose-leaf are the most common forms grown in aquaponic and hydroponic systems due to the rapid increase in weight and shorter time to maturity [3].

Plants require certain macro- and micronutrients to optimize growth, yield, and complete their lifecycle [5]. The primary macronutrients (nitrogen (N), phosphorous (P), and potassium (K)) are present in the greatest quantities within plants as compared to the secondary macronutrients (calcium (Ca), magnesium (Mg), and sulfur (S)). These nutrients are utilized in key functions in plant development that cannot be substituted by another element. Additionally, proper fertility management is an economical goal of growers to

maximize yield while minimizing inputs. Many of these nutrients result in a yield loss when deficient. In other cases, a decrease in secondary metabolites, such as anthocyanin and chlorophyll, resulting in a decrease in visual quality before a decrease in yield is observed [6]. Mineral nutrients, such as P, often play an indirect role through the promotion of photosynthesis and increase sugars used in secondary metabolite synthesis such as anthocyanin, which can increase pigment per unit of leaf weight as a result of limited leaf expansion [7].

The visual quality of fresh produce such as lettuce is one of the most important aspects considered by consumers [8]. Many aspects can impact the visual appearance of lettuce, one of which is suboptimal fertility rates of plant essential nutrients. One example is Mg deficiency. As the central atom of the chlorophyll molecule, a deficiency in Mg can result in decreased chlorophyll concentration without exhibiting visual Mg deficiency symptoms [9]. While these deficiencies may need to be in advanced stages before a decrease in plant biomass occurs, a decrease in plant pigments may occur, resulting in visually unappealing produce. Visual symptoms can be utilized as indicators of a specific mineral deficiency and have proved to be useful in assisting growers in performing fertility adjustments [10]. To properly diagnose nutrient deficiencies that result in visual deficiencies and recommend fertility adjustments, plant tissue analysis is essential. Plant tissue analysis was extensively utilized to evaluate the nutritional status of a crop and essential nutrient sufficiency levels, and to develop recommendations for fertilizer rates [11].

One of the most common nutrient deficiencies of lettuce is Ca deficiency. Calcium is an immobile element that is utilized in the formation and stabilization of cell walls [12]. Tipburn is a physiological disorder commonly associated with localized Ca deficiency occurring in newly developing foliage, and is often increased under high temperature, low humidity, high radiation, and high $CO_2$ [13–15]. Growers need to have adequate resources to ensure optimal Ca fertility rates and to understand the impact of suboptimal Ca uptake on growth even if visual nutrient deficiency symptoms do not appear.

Many other crops have been evaluated for nutrient deficiency impact on plant growth in hydroponic systems, including bok choy (*Brassica rapa var. Chinensis*) [6], carinata (*Brassica carinata*) [16], and cannabis (*Cannabis sativa*) [11]. However, these studies did not establish the impact of macronutrient fertility rates on uptake when altering a single element. Given the popularity of lettuce, other researchers published on the deficiency symptoms and nutrient concentrations at which deficiencies appear for several different lettuce cultivars [8,17]. Additionally, many researchers examined the impact of electrical conductivity (EC) on biomass and secondary metabolites [18–21]. Zanin et al. [18] reported that increased EC levels resulted in a reduction in yield and leaf nitrate concentrations; however, an increase in total phenolics was reported. Additionally, an increase in EC levels increased phosphorus (P), iron (Fe), manganese (Mn), and zinc (Zn) foliar concentrations in lettuce [19]. Additionally, nitrogen (N) rates were evaluated by comparing organic and nonorganic N sources, and it was determined that, when using the same N fertility rate, plants grown with organic fertilizers exhibited a lower N foliar concentration [22]. However, there is limited research on the impact of individual element fertility concentrations on plant growth or how fertility needs to vary throughout the production cycle.

Proper fertility is necessary to promote plant development and visually appealing crops. Providing nutrient fertility rates to promote maximal plant yield without resulting in luxury consumption should be an economic and sustainable priority for all growers. This study was conducted to examine the independent impact of each macroelement on plant growth and quality. The objectives are to provide growers with a total N, P, K, Ca, Mg, and S fertility program to optimize plant biomass production, to minimize fertilizer inputs, and to provide foliar nutrient concentrations associated with optimal growth.

## 2. Materials and Methods

*Lactuca sativa* 'Salanova Green' (Johnny's Select Seeds; Winslow, ME, USA) seeds were sown on 12 January 2021 into 276-count cell sheets (Oasis Horticubes, Oasis Grower

Solutions, Kent, OH, USA) and germinated in a mist bench for two weeks. 'Salanova Green' seedlings were transplanted into 12.7 cm diameter (1.29 L) plastic pots containing silica-sand (Millersville #2 (0.8 to 1.2 mm diameter) from Southern Products and Silica Co., Hoffman, NC, USA) on 26 January 2021. Plants were grown in a glass greenhouse at 18.3 and 15.5 °C ± 3.1°C days/nights in Raleigh, NC, USA (35 °N latitude). Immediately after transplanting, fertility treatments were initiated using an automated recirculating irrigation system that was constructed out of a 10.2 cm diameter PVC pipe (Charlotte Plastics, Charlotte, NC, USA). Detailed information about the formulation of the fertilizer treatments, salts used, and the (recirculating irrigation) system is given in Barnes et al. [23].

## 2.1. Fertility Treatments

Fertility macronutrient treatments were subdivided into six different concentrations (0, 8.3, 16.7, 33.3, 66.7, and 100%) of a modified Hoagland's solution [24] based on 150 mg·L$^{-1}$ N and following similar ratios (Table 1). While varying the concentration of desired macronutrients, all other nutrients were maintained constant among the treatments. All fertilizers were custom blends of the following individual technical grade salts (Fisher Scientific, Pittsburgh, PA): calcium nitrate tetrahydrate [$Ca(NO_3)_2 \cdot 4H_2O$], potassium nitrate ($KNO_3$), monopotassium phosphate ($KH_2PO_4$), potassium sulfate ($K_2SO_4$), magnesium sulfate heptahydrate ($MgSO_4 \cdot 7H_2O$), magnesium nitrate [$Mg(NO_3)_2$], monopotassium phosphate ($KH_2PO_4$), sodium phosphate heptahydrate ($NaH_2PO_4 \cdot 7H_2O$), iron chelate (Fe-DTPA), manganese chloride tetrahydrate ($MnCl_2 \cdot 4H_2O$), zinc chloride heptahydrate ($ZnCl_2 \cdot 7H_2O$), copper chloride dihydrate ($CuCl_2 \cdot 2H_2O$), boric acid ($H_3BO_3$), and sodium molybdate dihydrate ($Na_2MoO_4 \cdot 2H_2O$). Fertilization treatments began on the day of transplant.

**Table 1.** Calculations for modified Hoagland's solution were utilized to explore the impacts of varying macronutrients on the growth of *Lactuca sativa* 'Salanova Green'. Values indicate the adjusted fertility concentration provided from a modified Hoagland's solution with all elements held constant except the adjusted macroelement being investigated.

| Fertility Rate (%) [1] | 0.0 mg·L$^{-1}$ | 8.0 mg·L$^{-1}$ | 16.7 mg·L$^{-1}$ | 33.3 mg·L$^{-1}$ | 66.7 mg·L$^{-1}$ | 100.0 mg·L$^{-1}$ |
|---|---|---|---|---|---|---|
| Nitrogen (N) | 0.0 | 12.0 | 24.0 | 48.0 | 96.0 | 150.0 |
| Phosphorus (P) | 0.0 | 1.63 | 3.25 | 6.50 | 13.0 | 20.0 |
| Potassium (K) | 0.0 | 12.0 | 24.0 | 48.0 | 96.0 | 150.0 |
| Calcium (Ca) | 0.0 | 4.7 | 9.8 | 18.75 | 37.5 | 75.0 |
| Sulfur (S) | 0.0 | 2.5 | 5.0 | 10.0 | 20.0 | 40.0 |
| Magnesium (Mg) | 0.0 | 2.0 | 4.0 | 8.0 | 16.0 | 25.0 |
| | Micronutrient Fertility Rate (mg·L$^{-1}$) [2] | | | | | |
| All fertility rates | Fe | Mn | Cu | Zn | B | Mo |
| | 4.02 | 0.99 | 0.19 | 0.20 | 0.49 | 0.01 |

[1] These values are expressed as a percentage of the standard Hoagland's solution. For more detailed information on the system and fertility modifications, see Barnes et al. (2012) [23]. [2] Micronutrient fertility concentrations were held constant for all examined macronutrient fertility rates.

## 2.2. Plant Biomass and Foliar Samplings

At set intervals of three, six, and eight weeks after transplant, four representative plants from each nutrient rate were selected for sampling. The last data collection occurred on 23 March 2021, and the study was terminated. The entire shoots of four representative plants were sampled (n = 4) to evaluate the critical tissue concentration for each element. Collected shoots were washed in a solution of 0.5 M HCl for 1 min and then rinsed with deionized water. Upon sampling, the shoots were dried at 70 °C for 96 h, and the dry mass was weighed and recorded. After drying, leaf tissue was ground in a Foss Tecator Cyclotec™ 1093 sample mill (Analytical Instruments, LLC; Golden Valley, MN, USA; ≤0.5 mm sieve). The ground tissue was then placed in vials and analyzed for plant essential nutrients by the North Carolina Department of Agriculture and Consumer Services (NCDA) testing lab (Raleigh, NC, USA). Plant material (0.5 g) was digested in nitric acid (10 mL of $HNO_3$ at 15.6 N) in a

closed-vessel microwave digestion system for 30 min (MARS 6 Microwaves; Matthews, NC, USA). After microwave digestion, the plant material was diluted with 50 mL of deionized water and then vacuum-filtered through acid-washed paper (Laboratory Filtration Group; Houston, TX, USA). After dilution, plant mineral tissue concentration was determined with inductively coupled plasma-optical emission spectrometry (ICP-OES) (Spectro Arcos EOP; Mahwah, NJ, USA).

### 2.3. Statistical Analysis

Statistical analysis was conducted using SAS (version 9.4; SAS inst., Cary, NC, USA). Plant dry weight and leaf nutrient values were analyzed for differences within each data collection regarding the element's fertility rate as the explanatory variable using PROC GLM. Where the *F*-test was significant, LSD with a Tukey–Kramer adjustment ($p < 0.05$) was used to compare differences among means. Deviations in plant dry weight and leaf tissue values were calculated on a percentage basis from the controls (100% fertility rate). Regression models (linear, quadratic, or quadratic plateau) were compared, and the equation of best fit that resulted in the greatest statistical significance and the greatest $R^2$ values was selected.

## 3. Results and Discussion

### 3.1. Nitrogen

#### 3.1.1. Nitrogen Nutrient Deficiency Symptoms

Nitrogen deficiency symptoms were first expressed as stunted growth and a light green coloration of the foliage (Figure 1). As symptoms progressed, the middle and lower foliage turned chlorotic. In severe cases, necrosis formed on the lower leaves starting at the leaf margins and progressing to the midrib. Symptoms were observed on plants that received a fertility rate of 0.0, 12.0, or 24.0 mg·L$^{-1}$ N, and were first observed at Week 3, while plants that received a fertility rate of 48 mg·L$^{-1}$ N or greater did not develop visual deficiency symptoms at any time during the experiment.

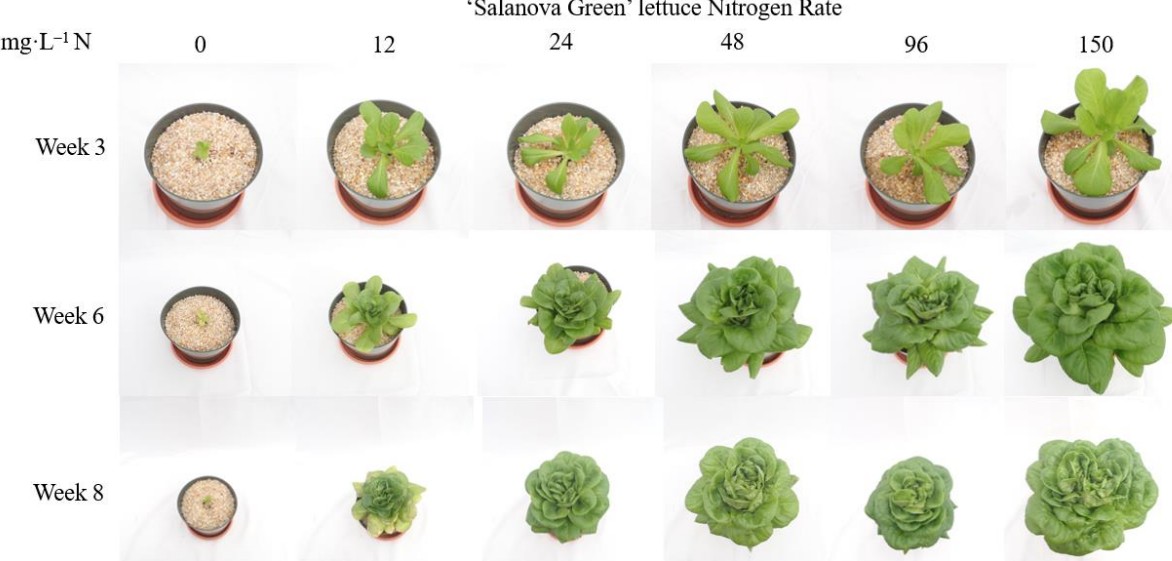

**Figure 1.** Visual impact of nitrogen (N) fertility rate on *Lactuca sativa* 'Salanova Green' at Weeks 3, 6, and 8.

#### 3.1.2. Nitrogen Leaf Tissue Accumulation and Biomass

At Week 3, plants that had received the highest N fertility rate of 150.0 mg·L$^{-1}$ N exhibited a 11.5× greater plant dry weight than that of plants that had received the lowest N fertility rate of 0.0 mg·L$^{-1}$ N (Table 2). At Week 6, plants that had received the highest N fertility rate of 150.0 mg·L$^{-1}$ N exhibited significantly greater plant dry mass

when compared to plants that had received the three lowest examined N fertility rates ($\leq$24.0 mg·L$^{-1}$ N) (Table 2). At Week 8, plants that received the highest N fertility rate (150.0 mg·L$^{-1}$ N) exhibited significantly greater plant dry mass when compared to all other examined N fertility rates (Table 2).

**Table 2.** Impact of nitrogen fertility rate on plant dry weight and nitrogen leaf tissue concentration on 'Salanova Green' lettuce.

| | 'Salanova Green' | | | | | | | |
|---|---|---|---|---|---|---|---|---|
| Nitrogen Fertility Rate (mg·L$^{-1}$) [1] | 0 | 12 | 24 | 48 | 96 | 150 | *p*-Value [2] | *Equation of Best Fit* |
| | Plant Dry Weight (g) | | | | | | | |
| Week 3 | 0.02 C | 0.11 BC | 0.14 AB | 0.19 AB | 0.18 AB | 0.23 A | *** | (DW) = 0.024 + 0.007X − 0.0001X$^2$; Xo = 52.78 |
| Week 6 | 0.02 C | 0.89 BC | 1.35 B | 3.35 A | 3.23 A | 4.03 A | *** | (DW) = 0.166 + 0.094X − 0.0006X$^2$; Xo = 81.80 |
| Week 8 | 0.05 D | 2.20 C | 3.28 C | 6.58 B | 5.15 B | 9.33 A | *** | (DW) = 1.603 + 0.051X; R$^2$ = 77.41 |
| | Nitrogen Leaf Tissue Nutrient Concentrations (%) | | | | | | | |
| Week 3 | 1.11 C | 3.54 B | 3.96 AB | 3.94 AB | 4.77 A | 4.79 A | *** | (N) = 1.321 + 0.206X − 0.0034X$^2$; Xo = 30.19 |
| Week 6 | 1.19 D | 1.63 D | 2.97 CD | 3.90 BC | 4.73 AB | 5.48 A | *** | (N) = 1.196 + 0.066X − 0.0003X$^2$; Xo = 124.54 |
| Week 8 | 1.31 C | 1.81 C | 3.39 B | 3.76 B | 5.48 A | 5.35 A | *** | (N) = 1.263 + 0.074X − 0.0003X$^2$; Xo = 111.40 |

[1] Values indicate the adjusted fertility rate provided from a modified Hoagland's solution with all elements held constant except the studied adjusted microelement. [2] *** indicates statistically significant differences between sample means based on the *F*-test (proc GLM) at $p \leq 0.001$. NS (not significant) indicates the *F*-test difference between sample means was $p > 0.05$. Values with the same letter indicate a lack of statistical significance, while values with different letters indicate statistically significant results.

At Week 8, when comparing the relationship between plant dry weight and N fertility rate, a linear increase was observed; however, when examining the relationship between N fertility rate and foliar N concentrations, a quadratic plateau was the equation of best fit (Table 2). At Weeks 6 and 8, a plateau in N foliar concentration was observed at 124.54 and 111.40 mg·L$^{-1}$ N, respectively (Table 2). However, no plateau was observed regarding plant dry weight at any of the examined N fertility rates at Week 8, which supports the use of 150 mg·L$^{-1}$ N to increase plant dry weight.

These results expand on those reported by Henry et al. [8], in which N deficiency was initially observed on 'Salanova Green' plants at 2.65% N. These data suggest that N deficiency is initially observed with a greater concentration of 3.96% N for three-week-old plants, and it declines to 2.97% N by Week 6. It also expands the optimal N foliar concentration to 4.79–5.48%.

*3.2. Phosphorus*

3.2.1. Phosphorus Nutrient Deficiency Symptoms

Phosphorus deficiency symptoms were initially observed as an overall stunting of the plant and a light green coloration of lower foliage (Figure 2) on plants that had received a P fertility rate of 0.00 mg·L$^{-1}$ P at Week 3, and 1.63 mg·L$^{-1}$ P at Week 6. As symptoms progressed, the upper foliage appeared dark green, and in severe cases, necrosis formed on the lower foliage. Additionally, P-deficient plants exhibited dark, dull green coloration when compared to the control plants.

3.2.2. Phosphorus Leaf Tissue Accumulation and Biomass

At Week 3, there was no discernable relationship between plant dry weight and P fertility rates (Table 3). However, at Weeks 6 and 8, quadratic plateau modeling provided the best fit. At Weeks 6 and 8, a Xo of 10.81 and 15.15 mg·L$^{-1}$ P, respectively, was observed and provided optimal plant growth (Table 3).

When examining the relationship between foliar P concentration and P fertility rate, quadratic plateau modeling exhibited the best fit. At Weeks 3, 6, and 8, plateaus were observed at 5.47, 14.76, and 9.86 mg·L$^{-1}$ P, respectively (Table 3). Maximizing harvest weight is the goal of lettuce production; thus, these results suggest a P fertility rate of 15.15 mg·L$^{-1}$ P for growers to maximize yield and provide adequate leaf tissue P concentrations for 'Salanova Green'. The modified Hoagland's solution provided by Domingues et al. [25]

utilized a P fertility rate of 21.7 mg·L$^{-1}$ P; however, this research suggests that growers could reduce P fertility by 30.1% in a similar system.

'Salanova Green' lettuce Phosphorus Rate

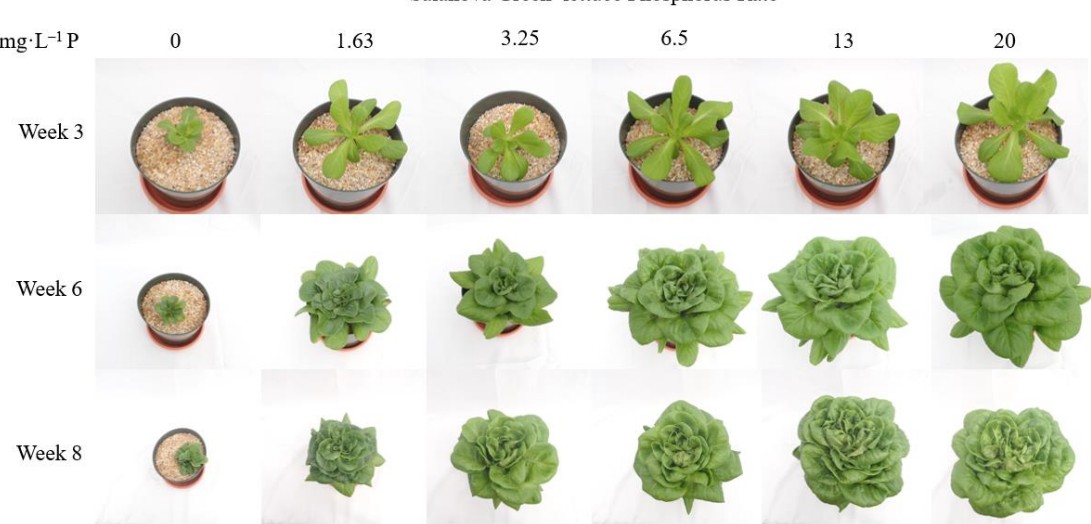

**Figure 2.** Visual impact of phosphorus (P) fertility rate on *Lactuca sativa* 'Salanova Green' at Weeks 3, 6, and 8.

**Table 3.** Impact of phosphorus fertility rate on plant dry weight and phosphorus leaf tissue concentration on 'Salanova Green' lettuce.

| Phosphorus Fertility Rate (mg·L$^{-1}$) [1] | 0 | 1.63 | 3.25 | 6.5 | 13 | 20 | *p*-Value [2] | *Equation of Best Fit* |
|---|---|---|---|---|---|---|---|---|
| 'Salanova Green' | | | | | | | | |
| Plant Dry Weight (g) | | | | | | | | |
| Week 3 | 0.08 A | 0.16 A | 0.25 A | 0.16A | 0.16 A | 0.23 A | NS | NS |
| Week 6 | 0.45 C | 1.99 B | 1.55 BC | 3.58 A | 3.50 A | 4.03 A | *** | (DW) = 0.555 + 0.559X − 0.028X$^2$; Xo = 10.81 |
| Week 8 | 0.43 B | 2.77 B | 2.93 B | 6.88 A | 9.28 A | 9.33 A | *** | (DW) = 0.408 + 1.187X − 0.039X$^2$; Xo = 15.15 |
| Phosphorus Leaf Tissue Nutrient Concentrations (%) | | | | | | | | |
| Week 3 | 0.34 C | 0.54 BC | 0.74 AB | 0.87 A | 0.86 A | 0.71 A | ** | (P) = 0.321 + 0.178X − 0.016X$^2$; Xo = 5.47 |
| Week 6 | 0.09 C | 0.17 C | 0.43 B | 0.43 B | 0.66 A | 0.70 A | *** | (P) = 0.094 + 0.789X − 0.003X$^2$; Xo = 14.76 |
| Week 8 | 0.12 C | 0.14 C | 0.44 B | 0.55 AB | 0.73 A | 0.49 B | *** | (P) = 0.073 + 0.1092X − 0.006X$^2$; Xo = 9.86 |

[1] Values indicate the adjusted fertility rate provided from a modified Hoagland's solution with all elements held constant except the adjusted microelement being studied. [2] ** or *** indicates statistically significant differences between sample means based on *F*-test (proc GLM) at $p \leq 0.01$, or $p \leq 0.001$, respectively. NS (not significant) indicates the *F*-test difference between sample means was $p > 0.05$. Values with the same letter indicate a lack of statistical significance, while values with different letters indicate statistically significant results.

Phosphorus deficiency symptoms were observed when plants exhibited a foliar P concentration of 0.09 to 0.17% at Weeks 6 and 8, which expands the P deficiency foliar concentration of 0.13% observed by Henry et al. [8]. At Week 3, P deficiency symptoms were observed on plants that had received a P fertility rate of 0.0 mg·L$^{-1}$ P and exhibited a P foliar concentration of 0.34% P, which was likely due to the concentration resulting from the limited plant growth three weeks after transplant.

### 3.3. Potassium

3.3.1. Potassium Nutrient Deficiency Symptoms

Potassium deficiency was initially observed as stunting on plants that had received K fertility rates of 0.0 and 12.0 mg·L$^{-1}$ K when compared to plants that had received the highest fertility rate of 150.0 mg·L$^{-1}$ K. Initial symptoms included downward leaf cupping, stunting, and marginal chlorosis of the lower foliage (Figure 3). As symptoms progressed, the necrosis spread to the center of the leaves, and lower leaves abscised from the plant.

Additionally, in advanced cases, K-deficient plants exhibited leaf distortion in the upper foliage. No deficiency symptoms were observed at $\geq$24 mg$\cdot$L$^{-1}$ K.

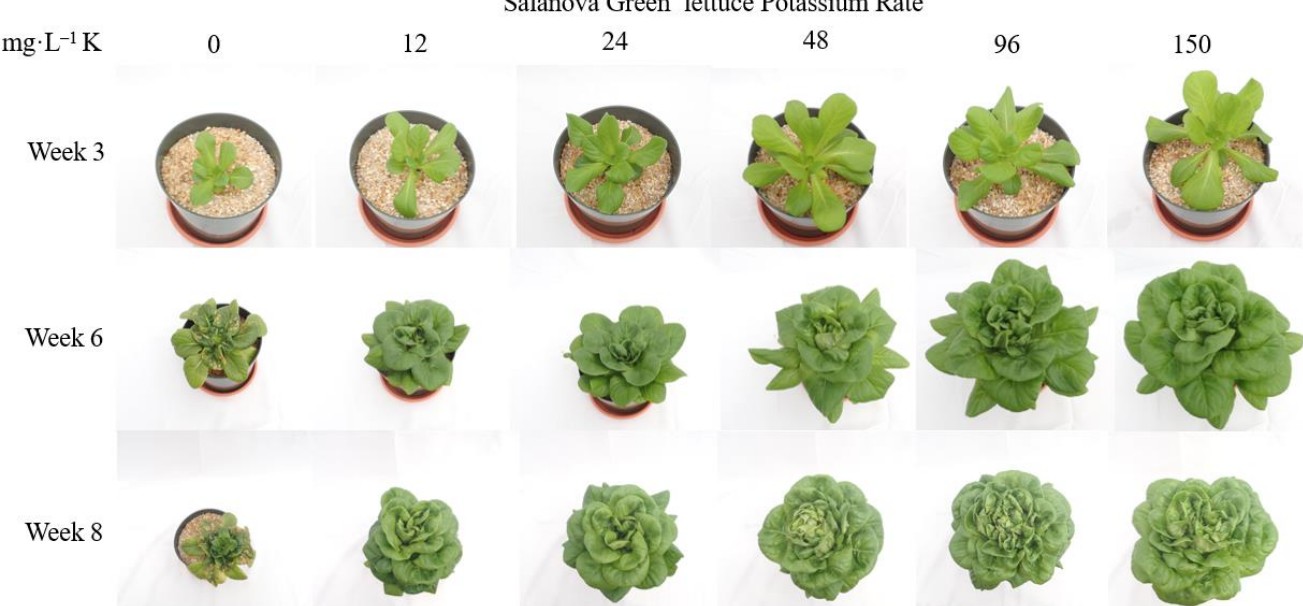

**Figure 3.** Visual impact of potassium (K) fertility rate on *Lactuca sativa* 'Salanova Green' at Weeks 3, 6, and 8.

### 3.3.2. Potassium Leaf Tissue Accumulation and Biomass

At Week 3, plants that had received the highest fertility rate of 150.0 mg$\cdot$L$^{-1}$ K had a 2$\times$ greater plant dry weight than that of plants receiving the lowest K fertility rate (0.0 mg$\cdot$L$^{-1}$ K) (Table 4). At Weeks 3, 6, and 8, the quadratic plateau model best represented the relationship between K fertility rate and plant dry weight. At Weeks 3, 6, and 8, a plateau was observed at rates of 74.67, 68.57, and 50.07 mg$\cdot$L$^{-1}$ K, respectively, and provided optimal plant growth (Table 4).

When examining the relationship between K fertility rate and K foliar concentration, a quadratic plateau was determined to be the model of best fit at Weeks 3 and 6, and a linear model provided the best fit at Week 8 (Table 4). At Weeks 3 and 6, plateaus were observed at 79.46 and 141.33 mg$\cdot$L$^{-1}$ K, respectively (Table 4). While the plateau for the relationship between K fertility rate and K foliar concentration was greater than that of the relationship between K fertility rate and plant dry weight, luxury consumption likely occurred. Growers aiming to maximize yield while still providing adequate K foliar concentrations should target a K fertility rate of 74.67 mg$\cdot$L$^{-1}$ K. When comparing this new recommended K fertility rate with that utilized by Domingues et al. [25] of 163.8 mg$\cdot$L$^{-1}$ K, our recommendation suggests that growers can decrease K fertility rate by half while still maintaining optimal plant growth.

At Week 3, plants that had received a K fertility rate of 0.0 mg$\cdot$L$^{-1}$ K had a foliar K concentration of 1.41% K, which was greater than the initial K deficiency foliar concentration of 0.79% K reported by Henry et al. [8]. This was likely due to the dilution effect, since the plants in this experiment were sampled earlier than those by Henry et al. (Table 4). At Weeks 6 and 8, deficiency symptoms were observed on plants that exhibited a K foliar concentration ranging from 0.39 to 1.03% K (Table 4), which expands the previously reported K deficiency concentration of 0.79% K by Henry et al. [8].

**Table 4.** Impact of potassium fertility rate on plant dry weight and potassium leaf tissue concentration on 'Salanova Green' lettuce.

| Potassium Fertility Rate (mg·L$^{-1}$) [1] | 0 | 12 | 24 | 48 | 96 | 150 | *p*-Value [2] | *Equation of Best Fit* |
|---|---|---|---|---|---|---|---|---|
| | | | | 'Salanova Green' | | | | |
| | | | | Plant Dry Weight (g) | | | | |
| Week 3 | 0.11 B | 0.13 AB | 0.14 AB | 0.21 A | 0.18 AB | 0.23 A | *** | (DW) = 0.098 + 0.003X − 0.00002X$^2$; Xo = 74.67 |
| Week 6 | 1.10 C | 2.30 BC | 2.45 ABC | 3.85 AB | 3.80 AB | 4.03 A | *** | (DW) = 1.157 + 0.081X − 0.0006$^2$; Xo = 68.57 |
| Week 8 | 0.97 D | 5.40 BC | 4.75 C | 7.25 AB | 7.28 AB | 9.33 A | *** | (DW) = 1.516 + 0.233X − 0.0023$^2$; Xo = 50.07 |
| | | | | Potassium Leaf Tissue Nutrient Concentrations (%) | | | | |
| Week 3 | 1.41 C | 3.02 B | 3.87 B | 6.45 A | 6.81 A | 7.21 A | *** | (K) = 1.333 + 0.144X − 0.0009X$^2$; Xo = 79.46 |
| Week 6 | 0.39 D | 1.03 D | 4.88 BC | 4.24 C | 7.11 AB | 8.28 A | *** | (K) = 0.669 + 0.106X − 0.00037X$^2$; Xo = 141.33 |
| Week 8 | 0.47 B | 0.98 B | 2.74 B | 2.66 B | 5.49 A | 7.07 A | *** | (K) = 0.553 + 0.061X − 0.0001X$^2$ |

[1] Values indicate the adjusted fertility rate provided from a modified Hoagland's solution with all elements held constant except the adjusted microelement being studied. [2] *** indicates statistically significant differences between sample means based on *F*-test (proc GLM) at $p \leq 0.001$. NS (not significant) indicates the *F*-test difference between sample means was $p > 0.05$. Values with the same letter indicate a lack of statistical significance, while values with different letters indicate statistically significant results.

### 3.4. Calcium

#### 3.4.1. Calcium Nutrient Deficiency Symptoms

Calcium nutrient deficiency symptoms were first observed three weeks after transplant at Ca fertility rates of 0.0, 4.69, or 9.38 mg·L$^{-1}$ Ca. Initially, symptoms were observed as an overall plant stunting and tip burn on the newly developing foliage. This tip burn resulted in the impacted foliage cupping downward (Figure 4). As symptoms progressed, necrotic spotting formed on the impacted foliage, and in severe cases, leaves abscised from the plant. At Weeks 6 and 8, isolated occurrences of tip burn were observed in the center of the heads on plants that had received Ca fertility rates of 18.75 and 37.5 mg·L$^{-1}$ Ca; however, this was not consistent across all replicates and was likely caused by environmental effects.

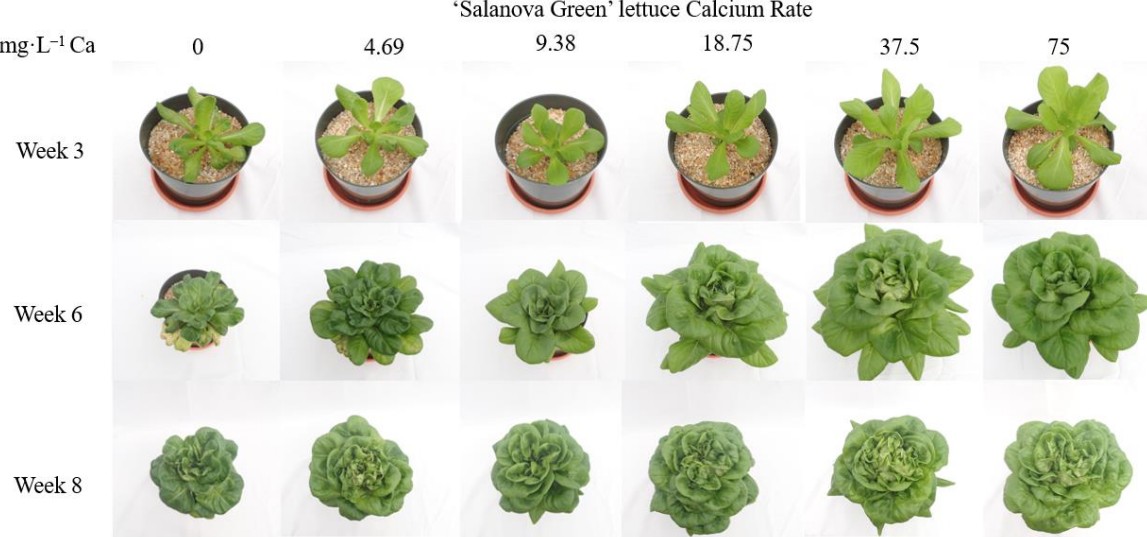

**Figure 4.** Visual impact of calcium (Ca) fertility rate on *Lactuca sativa* 'Salanova Green' at Weeks 3, 6, and 8.

#### 3.4.2. Calcium Leaf Tissue Accumulation and Biomass

At Weeks 3, 6, and 8, quadratic plateau models best represented the relationship between Ca fertility and plant dry weight. At these three intervals, plateaus of 68.91, 36.11, and 29.90 mg·L$^{-1}$ Ca, respectively, were observed (Table 5). Additionally, when examining the relationship between foliar Ca concentration and Ca fertility rate, a quadratic plateau model exhibited the best fit at Weeks 3 and 6. At Weeks 3 and 6, a plateau was observed at

23.34 and 50.28 mg·L$^{-1}$ Ca. However, at Week 8, the linear quadradic model exhibited the best fit (Table 5). For growers who want to maximize biomass production, a Ca fertility rate of 68.91 mg·L$^{-1}$ Ca allows for optimal biomass production while still providing adequate Ca foliar concentrations. This new Ca fertility rate recommendation is a 50.8% reduction in Ca fertility when compared to that utilized by Domingues et al. [25] of 140 mg·L$^{-1}$ Ca. Higher Ca rates are often provided to lettuce with the utilization of a CaNO$_3$ fertilizer source and the desire to avoid tip burn. Under optimal growing conditions, we found that plant dry weight at Week 8 was maximized with 29.90 mg·L$^{-1}$ Ca, but growers should use caution before adapting this lower rate. Calcium is a nonmobile element and is utilized by plants in many ways, including cell-wall and cell-membrane formation; however, uptake can be limited when plants are exposed to abiotic stress [26]. When plants are exposed to abiotic stress, one of the most common nutrient deficiencies in lettuce is tip burn as a result of localized Ca deficiency. Additional research is needed to determine if this lower rate of 68.91 mg·L$^{-1}$ Ca is achievable for lettuce under abiotic stress conditions.

**Table 5.** Impact of calcium fertility rate on plant dry weight and calcium leaf tissue concentration on 'Salanova Green' lettuce.

| | 'Salanova Green' | | | | | | | |
|---|---|---|---|---|---|---|---|---|
| Calcium Fertility Rate (mg·L$^{-1}$) [1] | 0.0 | 4.69 | 9.38 | 18.75 | 37.50 | 75.0 | *p*-Value [2] | *Equation of Best Fit* |
| | Plant Dry Weight (g) | | | | | | | |
| Week 3 | 0.14 B | 0.14 B | 0.12 B | 0.13 B | 0.21 A | 0.23 A | *** | (DW) = 0.117 + 0.003X − 0.00002X$^2$; Xo = 68.91 |
| Week 6 | 1.65 D | 2.15 BCD | 1.98 CD | 4.30 AB | 4.70 A | 4.03 ABC | *** | (DW) = 1.3823 + 0.168X − 0.00233X$^2$; Xo = 36.11 |
| Week 8 | 3.58 C | 5.68 B | 5.73 B | 9.25 A | 9.43 A | 9.33 A | *** | (DW) = 3.5274 + 0.397X − 0.00663X$^2$; Xo = 29.90 |
| | Calcium Leaf Tissue Nutrient Concentrations (%) | | | | | | | |
| Week 3 | 0.34 B | 0.38 B | 0.96 A | 1.07 A | 1.07 A | 1.17 A | *** | (Ca) = 1.840 + 0.073X − 0.00156X$^2$; Xo = 23.34 |
| Week 6 | 0.20 C | 0.26 C | 0.61 BC | 0.85 AB | 1.04 AB | 1.24 A | *** | (Ca) = 0.186 + 0.040X − 0.00040X$^2$; Xo = 50.28 |
| Week 8 | 0.26 C | 0.26 C | 0.58 BC | 0.67 BC | 0.86 B | 1.33 A | *** | (Ca) = 0.267 + 0.021X − 0.00009X$^2$ |

[1] Values indicate the adjusted fertility rate provided from a modified Hoagland's solution with all elements held constant except the adjusted microelement being studied. [2] *** indicates statistically significant differences between sample means based on *F*-test (proc GLM) at *p* ≤ 0.001. NS (not significant) indicates the *F*-test difference between sample means was *p* > 0.05. Values with the same letter indicate a lack of statistical significance, while values with different letters indicate statistically significant results.

*3.5. Sulfur*

3.5.1. Sulfur Nutrient Deficiency Symptoms

Sulfur deficiency symptoms were initially observed as a light green coloration throughout the entire plant and the stunting of plant growth. Symptoms only appeared on plants that had received an S fertility rate of 0.0 mg·L$^{-1}$ S and were initially observed at Week 6 (Figure 5).

3.5.2. Sulfur Leaf Tissue Accumulation and Biomass

While there was a limited visual impact of S deficiency, there was a significant impact on plant dry weight at all three sample points. At Weeks 3, 6, and 8, quadratic plateau models best represented the relationship between S fertility and plant dry weight. At these three intervals, plateaus of 20.29, 11.84, and 19.32 mg·L$^{-1}$ S, respectively, were observed and determined to provide optimal plant growth (Table 6). When examining the interaction between S fertility and S foliar concentrations, plateaus were observed at 6.99 and 3.30 mg·L$^{-1}$ S, respectively, at Weeks 6 and 8 (Table 6). This suggests that growers who are aiming to optimize plant biomass should target a fertility rate of 20.29 mg·L$^{-1}$ S early in the production cycle. When comparing this new recommended S fertility rate with the rate of 43.4 mg·L$^{-1}$ S utilized by Domingues et al. [25], this new recommendation suggests that a 53.2% decrease in S fertility rate would still result in optimal plant growth.

Plants that had received the lowest S fertility rate of 0.0 mg·L$^{-1}$ S exhibited significantly lower S foliar concentrations when compared to all other examined S fertility rates (Table 6). This suggests expanding the previously reported S deficiency foliar concentration of 0.09%

S by Henry et al. [8] to include 0.9 to 0.19% S. Plants that had received one of the five highest examined S fertility rates of 2.0, 4.0, 8.0, 16.0, and 25.0 mg·L$^{-1}$ S exhibited similar foliar leaf concentrations after eight weeks of growth (Table 6). With limited variation in S leaf tissue concentrations coupled with similar dry weights, this suggests that the recommended S fertility rate of 25.0 mg·L$^{-1}$ S can be lowered to 20.29 mg·L$^{-1}$ S. This recommended range would achieve optimal plant dry weight while not exhibiting foliar S deficiency symptoms.

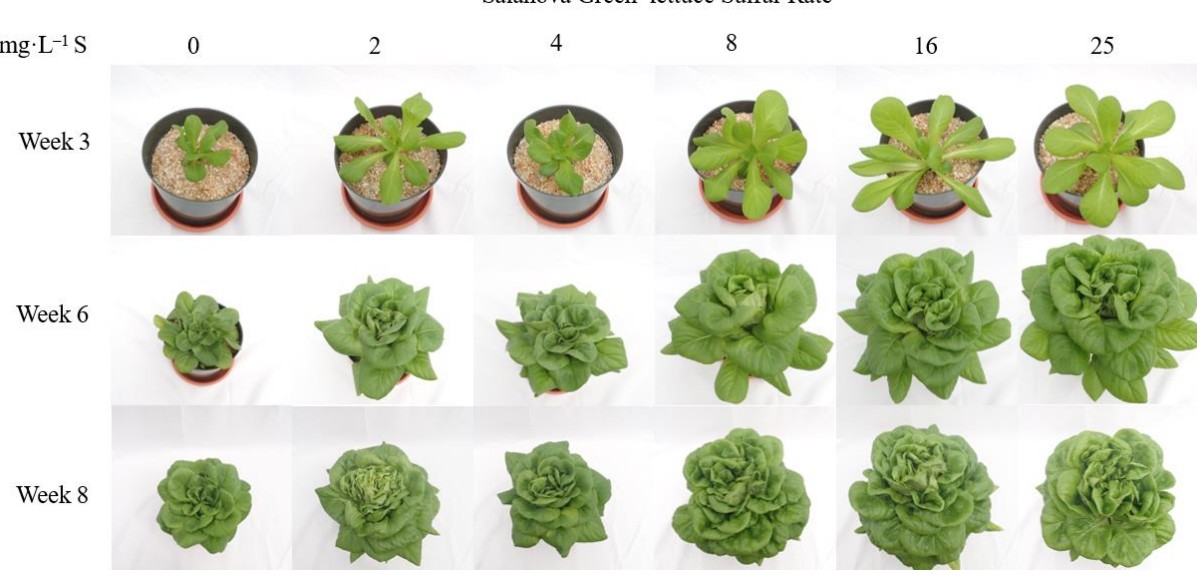

**Figure 5.** Visual impact of sulfur (S) fertility rate on *Lactuca sativa* 'Salanova Green' at Weeks 3, 6, and 8.

**Table 6.** Impact of sulfur fertility rate on plant dry weight and sulfur leaf tissue concentration on 'Salanova Green' lettuce.

| Sulfur Fertility Rate (mg·L$^{-1}$) [1] | **'Salanova Green'** | | | | | | | |
|---|---|---|---|---|---|---|---|---|
| | 0.0 | 2.0 | 4.0 | 8.0 | 16.0 | 25.0 | *p*-Value [2] | *Equation of Best Fit* |
| | *Plant Dry Weight (g)* | | | | | | | |
| Week 3 | 0.11 B | 0.16 AB | 0.17 AB | 0.19 AB | 0.25 AB | 0.30 A | * | (DW) = 0.120 + 0.013X − 0.00031X$^2$; Xo = 20.29 |
| Week 6 | 1.65 C | 3.60 AB | 2.73 BC | 4.35 A | 4.43 A | 4.55 A | *** | (DW) = 1.964 + 0.427X − 0.0180X$^2$; Xo = 11.84 |
| Week 8 | 4.40 C | 8.75 AB | 5.60 BC | 8.18 AB | 10.23 A | 9.70 A | *** | (DW) = 5.397 + 0.469X − 0.0121X$^2$; Xo = 19.32 |
| | *Sulfur Leaf Tissue Nutrient Concentrations (%)* | | | | | | | |
| Week 3 | 0.19 B | 0.25 A | 0.26 A | 0.28 A | 0.29 A | 0.28 A | *** | NS |
| Week 6 | 0.19 B | 0.25 A | 0.25 A | 0.28 A | 0.28 A | 0.29 A | *** | (S) = 0.197 + 0.025X − 0.0018X$^2$; Xo = 6.99 |
| Week 8 | 0.15 B | 0.26 A | 0.27 A | 0.28A | 0.29 A | 0.28 A | *** | (S) = 0.153 + 0.075X − 0.0114X$^2$; Xo = 3.30 |

[1] Values indicate the adjusted fertility rate provided from a modified Hoagland's solution with all elements held constant except the adjusted microelement being studied. [2] * or *** indicates statistically significant differences between sample means based on *F*-test (proc GLM) at $p \leq 0.05$, or $p \leq 0.001$, respectively. NS (not significant) indicates the *F*-test difference between sample means was $p > 0.05$. Values with the same letter indicate a lack of statistical significance, while values with different letters indicate statistically significant results.

### 3.6. Magnesium

3.6.1. Magnesium Nutrient Deficiency Symptoms

Magnesium deficiency symptoms were initially observed with plants that received an Mg fertility rate of 0.0 or 2.5 mg·L$^{-1}$ Mg at Week 6. Plants initially exhibited interveinal chlorosis of the lower foliage (Figure 6). As symptoms progressed, interveinal chlorosis was observed in the middle and upper foliage, and the interveinal chlorosis turned necrotic. In severe cases, the impacted foliage turned necrotic and abscised. Additionally, deficiency symptoms were never observed on plants that received an Mg fertility rate $\geq 5.0$ mg·L$^{-1}$ Mg.

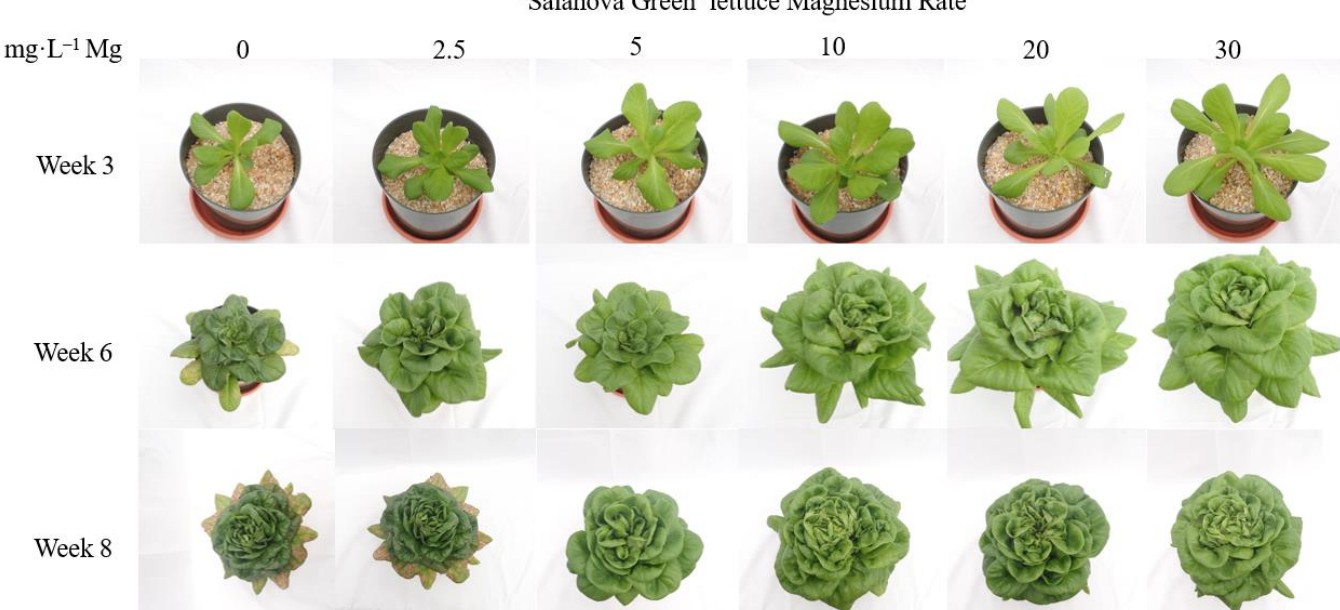

**Figure 6.** Visual impact of magnesium (Mg) fertility rate on *Lactuca sativa* 'Salanova Green' at Weeks 3, 6, and 8.

3.6.2. Magnesium Leaf Tissue Accumulation and Biomass

At Week 3, Mg fertility did not exhibit a significant impact on plant dry weight (Table 7). At Weeks 6 and 8, the relationship between Mg fertility rate and plant dry weight was optimized with the quadratic plateau models of 22.08 and 11.73 mg·L$^{-1}$ Mg (Table 7). However, when examining the relationship between Mg fertility rate and Mg foliar concentration at Weeks 3 and 8, the model of best fit was a quadratic model in which a plateau was not observed (Table 7). This suggests that, for growers aiming to maximize plant biomass, an Mg fertility rate of 22.08 mg·L$^{-1}$ Mg allows for optimal biomass production while still providing the plant with adequate Mg foliar concentrations. The modified Hoagland's solution provided by Domingues et al. [25] utilized an Mg fertility rate of 33.6 mg·L$^{-1}$ Mg. This research suggests that growers could reduce Mg fertility by 34.3% in a hydroponic system.

**Table 7.** Impact of magnesium fertility rate on plant dry weight and magnesium leaf tissue concentration on 'Salanova Green' lettuce.

| 'Salanova Green' | | | | | | | | |
|---|---|---|---|---|---|---|---|---|
| Magnesium Fertility Rate (mg·L$^{-1}$) [1] | 0.0 | 2.5 | 5.0 | 10.0 | 20.0 | 30.0 | *p*-Value [2] | *Equation of Best Fit* |
| Plant Dry Weight (g) | | | | | | | | |
| Week 3 | 0.14 A | 0.12 A | 0.17 A | 0.16 A | 0.19 A | 0.18 A | NS | NS |
| Week 6 | 1.18 C | 2.65 BC | 2.20 BC | 3.90 AB | 4.53 A | 4.83 A | *** | (DW) = 1.369 + 0.303X − 0.0069X$^2$; Xo = 22.08 |
| Week 8 | 2.27 D | 5.90 BC | 5.20 C | 9.10 A | 7.88 AB | 8.30 A | *** | (DW) = 2.584 + 0.979X − 0.0417X$^2$; Xo = 11.73 |
| Magnesium Leaf Tissue Nutrient Concentrations (%) | | | | | | | | |
| Week 3 | 0.26 D | 0.30 D | 0.38 C | 0.46 B | 0.43 BC | 0.56 A | *** | (Mg) = 0.278 + 0.015X − 0.0022X$^2$ |
| Week 6 | 0.10 D | 0.15 CD | 0.19 C | 0.35 B | 0.40 AB | 0.45 A | *** | (Mg) = 0.089 + 0.027X − 0.0005X$^2$; Xo = 25.85 |
| Week 8 | 0.10 D | 0.19 CD | 0.29 C | 0.46 B | 0.55 B | 0.71 A | *** | (Mg) = 0.112 + 0.035X − 0.0005X$^2$ |

[1] Values indicate the adjusted fertility rate provided from a modified Hoagland's solution with all elements held constant except the adjusted microelement being studied. [2] *** indicates statistically significant differences between sample means based on *F*-test (proc GLM) at $p \leq 0.001$. NS (not significant) indicates the *F*-test difference between sample means was $p > 0.05$. Values with the same letter indicate a lack of statistical significance, while values with different letters indicate statistically significant results.

## 4. Conclusions

Macronutrient fertility requirements varied by the element and growth stages of *Lactuca sativa* 'Salanova Green'. Biomass production and leaf tissue concentrations can be used as guidelines to help in informing other researchers and growers regarding hydroponic fertility and plant uptake requirements. These results indicate that, for some elements such as N, a linear increase in plant dry weight regarding increasing N fertility after eight weeks of plant growth, yet other elements such as Mg exhibited a plateau of plant biomass regarding increasing Mg fertility; however, an increase in foliar concentration was observed, suggesting luxury uptake by the plants. These results suggest the N fertility rates had the greatest impact on plant dry mass when all other elements were held constant. Additionally, other elements such as S may exhibit limited to no visual symptoms, but when deficient, a significant decrease in dry weight occurs. The results of this study suggest that a nutrient solution with nutrient concentrations (mg·L$^{-1}$): N, 150; P, 15.15; K, 74.67; Ca, 68.91; S, 20.29; Mg, 22.08 would provide optimal plant growth and avoid visual nutrient deficiency symptoms for 'Salanova Green' lettuce. These new fertility rate recommendations are significantly lower than those previously provided, allowing for growers to decrease costly inputs and negative environmental impacts.

**Author Contributions:** Conceptualization, P.V, M.S.B. and B.W.; methodology, P.V, M.S.B., P.P., S.Y., K.H. and B.W.; software, P.V., M.S.B., P.P., S.Y. and B.W.; validation, P.V., M.S.B., P.P., S.Y., K.H. and B.W.; formal analysis, P.V., P.P., K.H. and B.W.; investigation, P.V., M.S.B., P.P., S.Y., K.H. and B.W.; resources, S.Y., K.H. and B.W.; data curation, P.V., M.S.B., P.P., S.Y., K.H. and B.W.; writing—original draft preparation, P.V.; writing—review and editing P.V., M.S.B., P.P., S.Y., K.H. and B.W.; visualization, P.V. and B.W. supervision, S.Y. and B.W.; project administration, B.W.; All authors have read and agreed to the published version of the manuscript.

**Funding:** This research received no external funding.

**Institutional Review Board Statement:** Not Applicable.

**Informed Consent Statement:** Not Applicable.

**Data Availability Statement:** Not applicable.

**Conflicts of Interest:** The authors declare no conflict of interest.

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
