# Peer review of "Impact of Macronutrient Fertility on Mineral Uptake and Growth of Lactuca sativa ‘Salanova Green’ in a Hydroponic System"

_horticulturae, doi:10.3390/horticulturae8111075_

Round 1
Reviewer 1 Report
Dear authors, congratulations for this great work that is of relevance. It shows interesting data for the field of hydroponics.

Author Response
Dear MDPI Horticulturae Reviewers,
Thank you for the feedback on our submission titled: “Impact of Macronutrient Fertility on Mineral Uptake and Growth of Lactuca sativa ‘Salanova Green’ in a hydroponic system”. We have enhanced the manuscript following the advice given from the reviewers below.
Reviewer One:
Thank you for the feedback.
Reviewer 2 Report
The manuscript title “Impact of Macronutrient Fertility on Mineral Uptake and Growth of Lactuca sativa ‘Salanova Green’ in a Hydroponic System” needs significant improvements and some scientific questions which need to be addressed clearly. I have some comments and suggestions for authors:
1- Please give a background (or problem which is needed to be solved), what were the scientific gap/ questions (first 2-3 lines) in the abstract.
2- Why only plant biomass parameter was studied?? Why other foliar parameters, such as leaf secondary metabolites, anthocyanin and chlorophyll etc., what was the impact on different nutrients on root? No root parameter was studied? Why? Root is very important for mineral uptake.
3- Only 2-3 references of hydroponic was added in whole manuscript? Add more details of hydroponic from other horticultural crops and their impact on plant biomass?
4- Root has significant impact on the leaf development and growth! What the effect of different minerals on root length, root dry mass, root fresh weight etc. root to shoot ratio….? I strongly suggest authors to add some more parameters to enhance the scientific worth of this research.
5- Introduction and discussion also needs significant improvements, add details of Macronutrient Fertility on Mineral Uptake under hydroponic system. Do not only write the mineral functions in plants.
6- Line 379: Why you have added references in conclusion section. Please move all the references in the discussion section. Based on your research give conclusion and write some future directions etc.
7- How you research was different from (Sublett et al. 2018)? the first reference that you have cited in the manuscript.
Author Response
Dear MDPI Horticulturae Reviewers,
Thank you for the feedback on our submission titled: “Impact of Macronutrient Fertility on Mineral Uptake and Growth of Lactuca sativa ‘Salanova Green’ in a hydroponic system”. We have enhanced the manuscript following the advice given from the reviewers below.
Reviewer One:
Thank you for the feedback.
Reviewer Two:
Thank you for feedback, we have enhanced the article following your comments addressed below
- Background Info. We have enhanced addressing the background problem and gap in the literature in the first 2-3 lines by adding the statement: “While other fertilizer rate work has been conducted on lettuce, the impact of each element has not been evaluated independently or determining adequate foliar tissue concentrations when all nutrients are plant available.” Hopefully this clarifies our objectives better.
- Other Parameters. Thank you for the feedback on this suggestion. While we did not measure other parameters such as secondary metabolites or root growth, we elected to focus on the shoot growth which is the economical aspect of the crop used by greenhouse growers and the corresponding foliar concentrations of the shoot growth. We elected to focus on these two parameters for the following reasons: plant biomass production is the main focus of yield for hydroponic lettuce and is correlated with many nutrient deficiencies of stunting. We also mainly focus on the foliar nutrient concentrations since this is what growers would utilize when determining nutrient deficiencies. With these two parameters being the largest tools for diagnosing fertility problems and establishing fertility rates we elected not to measure secondary metabolites. In regard to the comment of why we did not measure root parameters, we have observed that in the system utilized the impact of nutrient fertility on shoot growth is a direct correlation to what is occurring with the roots. If shoot growth is lacking, then root growth is also. In addition, the largest negative impacts occur with –Ca or –B. Growers do not sample the roots for nutrient analysis, only the foliage, so we elected to focus the parameters of our research on what is standard protocol for determine nutrient concentrations within a plant.
- Additional References. We have enhanced the article with additional information of other crops impact on biomass in regard to nutrient deficiencies when grown in a hydroponic system.
- Root Assessment. Thank you for pointing out this aspect for future work, please see comment #2 addressing similar comments. We believe that while root growth has an important role in plant shoot growth, the shoot growth is a direct reflection of what is occurring with the roots. Measuring roots or nutrient analysis of the roots is not a standard protocol for determining impact of nutrients on plant growth. Most literature measures above ground biomass as well as foliar nutrient concentrations which were measured in this study.
- Introduction and Discussion. We have enhanced the article with additional information regarding macronutrient fertility on plant uptake in hydroponic systems.
- Line 379. We have moved references from the conclusion and into the discussion.
- Sublett Research. This research varies greatly from Sublett et al. 2018. In their research the authors evaluated the impact of only EC rates on plant growth and above ground biomass. EC is a combined measurement of all fertilizer salts in total. In addition, their concentrations started with 200 mg/L of N and then increased. In our current research we were evaluating the impact of each element (N, P, K, Ca, S, and Mg) independently. Thus we went from 150 mg/L N and lower to 0. Our research allowed us to determine the impact of each element rates when all other elements were adequate, which is not considered in Sublett et al. 2018. Our goal was to determine concentrations of each of the 6 elements that were both economically and ecologically sound practices.
Round 2
Reviewer 2 Report
The authors revised the MS as suggested!